# ZnO Tips Dotted with Au Nanoparticles—Advanced SERS Determination of Trace Nicotine

**DOI:** 10.3390/bios11110465

**Published:** 2021-11-19

**Authors:** Jiaying Cao, Yan Zhai, Wanxin Tang, Xiaoyu Guo, Ying Wen, Haifeng Yang

**Affiliations:** The Education Ministry Key Lab of Resource Chemistry, College of Chemistry and Materials Science, Shanghai Normal University, Shanghai 200234, China; 1000479846@smail.shnu.edu.cn (J.C.); 1000459032@smail.shnu.edu.cn (Y.Z.); gxy2012@shnu.edu.cn (X.G.); ying.wen@shnu.edu.cn (Y.W.)

**Keywords:** SERS, ZnO tips, nicotine, AuNPs

## Abstract

Long-term exposure to nicotine causes a variety of human diseases, such as lung damage/adenocarcinoma, nausea and vomiting, headache, incontinence and heart failure. In this work, as a surface-enhanced Raman scattering (SERS) substrate, zinc oxide (ZnO) tips decorated with gold nanoparticles (AuNPs) are fabricated and designated as ZnO/Au. Taking advantage of the synergistic effect of a ZnO semiconductor with morphology of tips and AuNPs, the ZnO/Au-based SERS assay for nicotine demonstrates high sensitivity and the limit of detection 8.9 × 10^−12^ mol/L is reached, as well as the corresponding linear dynamic detection range of 10^−10^–10^−6^ mol/L. Additionally, the signal reproducibility offered by the SERS substrate could realize the reliable determination of trace nicotine in saliva.

## 1. Introduction

Nicotine, as one of pyridine alkaloids mainly existing in tobacco and cigarette smoke [1,2], facilitates neurotransmitter release (dopamine and others), deriving pleasure, mood modulation, stimulation of the central nervous system, cholinergic nerves, and sympathetic nerves of smokers [3]. In addition, nicotine is often used as a variety of nicotine replacement therapies (NRT), such as nicotine transdermal patches, nasal sprays, inhalers, sublingual tablets, and lozenges [4]. However, overexposure to nicotine causes a variety of human diseases, including lung damage/adenocarcinoma, nausea and vomiting, headache, incontinence and heart failure [5,6,7,8], which is in close connection with mortality rate based on published data in literature [9], and nicotine has been considered as one of the most highly toxic alkaloids [10]. E-cigarettes have boomed in the smoker market because a nicotine-free e-liquid (cigarette oil) with a specific odor (containing flavoring or coloring agents, vegetable glycerin, propylene glycol, etc.) is used [11]. E-liquids containing nicotine have been banned in many countries including the Taiwan region [12]. However, occasionally, nicotine is detectable in e-liquids, leading to inadvertent nicotine inhalation [13,14,15,16]. This calls for on-market detection of nicotine in the commercial products.

To date, fluorescence [17], liquid chromatography/mass spectrometry (LC-MS) [18], gas chromatography/mass spectrometry (GC-MS) [19], electrochemistry [20,21,22], electro-chemiluminescence [23], flow injection analysis [24] and high-performance liquid chromatography (HPLC) [25,26] have been explored as nicotine assays. As an example, GC-MS as a standard method for detecting e-liquid has been adopted in the laboratory [27]. The requirement of expensive instruments and skilled operators as well as the low-throughput and cumbersome sample pretreatment procedures make laboratory-based analysis technique inconvenient for onsite inspection aims [28]. Therefore, it is urgently necessary to develop a highly sensitive, onsite and rapid protocol for the detection of nicotine.

A portable infrared meter has been used for field-testing, but its poor sensitivity has failed to detect nicotine present in low amounts in the products or in the liquid samples [29]. In recent years, surface-enhanced Raman scattering (SERS) spectroscopy for onsite detection applications has aroused attention [30,31,32,33] because of its high Raman signal enhancement and the fast development of portable Raman systems. Especially, some stable SERS materials for the detection of nicotine have been reported, such as thiol-terminated molecular imprint microspheres [34], colloids [35,36,37], and Au-nanoparticle-coated paper [38] but the sample pretreatment process is still complicated. We reported that the gold-dotted magnetic composite was prepared and used to detect nicotine in saliva by using a magnetically optimized SERS strategy, but the quantitatively analytic performance needs further improvement [39].

Zinc oxide (ZnO), due to its good biocompatibility and high resistance to environmental pH or temperature changes, has been used to prepare semiconductor/noble metal-composite substrates [40], which show both electromagnetic enhancement and charge transfer effects, thus improving SERS’ detection sensitivity [41,42]. In this work, ZnO with abundant tips was synthesized and then loaded with dense gold nanoparticles into the gaps of tips (designated as ZnO/Au), creating a larger surface area for capturing target molecules via chemical interaction and improving Raman scattering enhancement due to the presence of numerous hotspots. As a result, ZnO/Au-based SERS method could be used to determine trace nicotine in saliva.

## 2. Experimental Section

### 2.1. Materials

Sodium hydroxide (NaOH), Trisodium citrate, nicotine and zinc acetate (Zn (CH_3_COO)_2_) were purchased from Sigma-Aldrich (St. Louis, MO, USA). HAuCl_4_·4H_2_O and ethanol were bought from National Pharmaceutical Chemical Reagent (Shanghai, China). Rhodamine 6G (R6G) was obtained from Adamas Reagent. All chemicals were of analytical grade. Ultrapure water (18.2 MΩ cm) was used throughout all experiments. Glassware was embathed in aqua regia and then thoroughly rinsed with ultrapure water.

### 2.2. Instrumentation

UV-vis spectra were collected by a UV-vis spectrophotometer (Shimadzu, Kyoto, Japan, UV-1800). Raman experiments were carried out by using a Dilor confocal laser Raman system (French, SuperLabRam II) equipped with a 633 nm He-Ne laser and semiconductor-cooled CCD detector. A field-emission scanning electron microscope (SEM, JEOL6380LV) was used to observe morphologies of materials.

### 2.3. Synthesis of ZnO/Au

Under the optimum conditions, ZnO/Au was synthesized by a hydrothermal method. In brief, 0.2 g of Zn(CH_3_COO)_2_ was dispersed in 70 mL ultrapure water by ultrasonication for 30 min. With sonication, 10 mL NaOH (2 mol/L) was added into the Zn(CH_3_COO)_2_ solution under constant stirring, and then transferred to an autoclave. The reaction temperature and time were set at 120 °C and 20 h, respectively. After cooling to room temperature naturally, samples were washed with ultrapure water several times to remove residual ions and molecules, and dried at 70 °C under vacuum. The prepared ZnO of about 0.015 g was dissolved in 25 mL ultrapure water and heated until slightly boiling while stirring. Then, 1 mL of 10^−3^ mol/L HAuCl_4_ solution and 2 mL of 1% trisodium citrate were injected under stirring for 30 min until the solution turned dark purple, to obtain ZnO tips/AuNPs successfully. With the same protocols, 1 and 2.5 mL of 1% trisodium citrate were added to prepare the ZnO dotted with different sizes of AuNPs, named as ZnO/Au_1_ and ZnO/Au_3_.

### 2.4. SERS Measurement

For SERS detection, the solution containing analyte was mixed with ZnO/Au composite suspension by using a volume ratio of 1:1. The Raman test was conducted by using a 633 nm laser line with power 5 mW and collection time of 3 s with 2 accumulations. For each substrate, 5 points in the coffee ring area were randomly selected for evaluating the relative standard derivation of detections.

## 3. Results and Discussion

### 3.1. Characterization of ZnO/Au

The three-dimensional (3D) ZnO/Au prepared by hydrothermal method is given in Figure 1. In Figure 1A,B, the scanning electron microscopic (SEM) images show that the obtained 3D ZnO nanoparticles with regular tips have uniform morphology with a diameter of approximately 6 μm. For ZnO/Au, as shown in Figure 1C,D, the gold nanoparticles were evenly and densely distributed on the surface of ZnO tips, which is beneficial to formation of the nanogaps between the AuNPs, generating “hotspots” which contribute to strong electromagnetic enhancement on Raman scattering of target molecules [43,44].

In Figure 2, the EDS mapping further confirms the presence of AuNPs in the composite substrate, and clearly, AuNPs are evenly distributed throughout the ZnO system.

### 3.2. Optimization for Preparation of ZnO/Au

The morphologic effect of ZnO on the optical properties of semiconductor was investigated. Therefore, tips-like ZnO and litchi-like ZnO (Figure 3) were synthesized simultaneously. In Figure 4, compared with litchi-like ZnO, the UV-vis spectrum of tips-like ZnO has strong absorption in the range of 400–800 nm. Therefore, tips-like ZnO was picked to prepare the composite ZnO/Au, which increased the adsorption of the laser line at 633 nm.

The SERS performance of gold nanoparticles of different sizes reduced on ZnO has been compared by optimizing the volume (1, 2, and 2.5 mL) of reducing agent (1% trisodium citrate solution) added, as seen in Appendix A. Their UV-vis spectra are presented in Appendix A and the surface plasmon resonance bands from AuNPs are clearly visible in the region from 500 to 600 nm. The diameter of AuNPs was evaluated according to the equation in the Appendix A, and the diameters of AuNPs are estimated at about 71.5, 43 and 15 nm, respectively. In Appendix A, in the case of R6G (10^−7^ mol/L solution) as the Raman probe, the SERS signal recorded on ZnO Tips/AuNPs has the greatest intensity. This might be due to the SPR band from AuNPs in ZnO/Au matching the excitation line of 633 nm and the suitable aggregation of AuNPs forming the hotspots at the surface of ZnO tips.

### 3.3. SERS Performance of ZnO/Au

It can be seen from Appendix A that the SERS peaks of R6G could be still observed when the concentration is even as low as 1 × 10^−11^ mol/L, confirming the superior enhancement effect of ZnO/Au. In Appendix A, a linear concentration dynamic range of R6G is located between 10^−11^–10^−6^ mol/L, with a correlation coefficient (R^2^) of 0.9624. The Raman enhancement factor (EF) was evaluated according to the equation in the Appendix A, and the EF of 1.07 × 10^8^ was reached.

For checking the enhancement effect of ZnO tips, SERS spectra of R6G were acquired on ZnO Tips/Au and pure AuNPs, respectively. As clearly seen in Figure 5, SERS’ effect for R6G recorded on ZnO tips/Au is much greater than that of gold nanoparticles. It depicts that the ZnO tips improve the SPR enhancement effect of AuNPs through effective aggregation of nanoparticles and mutual interaction of ZnO and AuNPs to optimize the electronic structure of composite SERS substrate. In addition, the chemical enhancement effect of ZnO also contributes to the SERS detection sensitivity. The reason is that the work function of ZnO (5.2 eV) is larger than that of Au (5.1 eV) and the Fermi energy level of ZnO is lower than that of Au, resulting in electron transfer from AuNPs to ZnO that occurs until their level of Fermi energy attains equilibration. As a result, it is beneficial to the improvement of the chemical interaction between target molecules and ZnO. Simultaneously, it is conducive to more molecules closely approaching the vicinity of the hotspot region, further providing an enhancement effect.

For investigating the signal reproducibility of ZnO tips/Au composite material, three different substrates were selected for examining preparation reproducibility. For each substrate, five points in the coffee ring area were randomly selected for evaluating the relative standard derivation of detections (Figure 6A). The statistic bars based on the Raman intensity at 613 cm^−1^ are presented in able 6B, and the computed relative standard derivation (RSD) is 8.92%, indicating that the ZnO/Au substrate has good uniformity and acceptable detection reproducibility.

### 3.4. SERS Detection of Nicotine

The concentration-dependent SERS spectra of nicotine in aqueous solutions were obtained by using ZnO tips/Au composite material. In Figure 7A, the SERS intensity of the nicotine increases as the nicotine concentration increases. The SERS band at 1589 cm^−1^ corresponding to the stretching of the pyridine ring of nicotine was selected for quantitative analysis according to the previously reported work [45]. Compared with the normal Raman bands [45], the SERS bands of nicotine recorded on ZnO tips/Au substate experienced blue-shifts, which hints at the adsorption of nicotine on the substate surface in physical and chemical fashions. In Figure 7B, a linear concentration dynamic range is located from 10^−10^–10^−6^ mol/L (Based on the corresponding regression equation: y = 4524.33214x + 46,355.33316, where x is the concentration of nicotine and y represents the normalized Raman intensity at 1589 cm^−1^, a correlation coefficient (R^2^) = 0.9657). The limit of detection (LOD, S/N = 3) of 8.9 × 10^−12^ mol/L and limit of quantitation (LOQ, S/N = 10) [46,47,48] of 2.9 × 10^−11^ mol/L could be achieved, due to the composite ZnO tips/Au possessing a larger surface area with physical adsorption capability. Furthermore, the interaction of ZnO and the pyridine moiety of nicotine allows the target species to reach the hotspot region [49,50]. Meanwhile, the SERS bands at 1031 cm^−1^ were also used for quantitative analysis and a linear concentration dynamic range from 10^−10^–10^−6^ mol/L with a correlation coefficient (R^2^) of 0.9663 could be reached.

For real applications, ZnO tips/Au SERS spectra of nicotine with different amounts spiked into saliva samples were acquired, and the peak at 1589 cm^−1^ due to easy observation without any spectral interference was chosen for determination of nicotine. The assignments of the main SERS bands for nicotine are tabulated in Appendix A. As presented in Figure 8, clearly, the nicotine level in saliva could be detected at as low as 1 × 10^−10^ mol/L. As tabulated in Table 1, the RSDs are in the range of 6.86–8.43%, and the reasonable recoveries are in the range from 96.14 to 100.75%, remarking the robustness of the ZnO-tips/Au-based SERS assay for detection of nicotine in saliva.

Herein, the reported methods for determining nicotine such as high-performance liquid chromatography, electrochemistry, fluorescence, and other methods, were summarized in Table 2. Clearly, the proposed ZnO-tips/Au-based SERS assay exhibits superior sensitivity and reasonable quantitative analysis performance due to interaction between AuNPs and ZnO tips improving the SPR effect. In addition, the ZnO tips/Au substrate demonstrates good stability under ambient conditions, which is beneficial to real applications.

## 4. Conclusions

In summary, the ZnO tips/Au composite was successfully fabricated through a hydrothermal synthesis method. ZnO tips/Au as a SERS substrate showed high sensitivity and signal reproducibility. The amplification of the Raman signal for nicotine should be attributed to the optimal SPR field (hotspots), and the promoted adsorption of target molecules by ZnO tips. The ZnO-tips/Au-based SERS assay was used to directly determine nicotine, and a satisfactory linearity for the concentration dynamic range of 10^−10^–10^−6^ mol/L (R^2^ = 0.9657) was reached, and the LOD was 8.9 × 10^−12^ mol/L, which meets the requirements of national standards. As a practical application, the concentration of nicotine in saliva as low as 1 × 10^−10^ mol/L was detectable, and the detection recovery was located within the accepted degree. It is our view that an ZnO-tips/Au-based SERS assay could be advanced to realize the inspection of nicotine abuse in market-available products.

## Data Availability

All data are contained within the article.

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
