# Peer review of "ZnO Tips Dotted with Au Nanoparticles—Advanced SERS Determination of Trace Nicotine"

_biosensors, 2021, doi:10.3390/bios11110465_

Round 1
Reviewer 1 Report
In this work, ZnO tips decorated with AuNPs (ZnO-Tips/AuNPs) are fabricated to detect trace nicotine in saliva with sensitivity and low detection limit. In general, the substrate design is not special for such detection; a lot of nano-structured substrates in the literature are competent to realize the detection with similar capability. Therefore, in considering with novelty and significance, the paper is not recommended.
The following questions are particularly addressed for your further submission.
- The so-called “hot-spots” was not well described. How the target species can reach the hot spots and form the SERS mechanism on the ZnO-Tips/AuNPs surface?
- The R6G molecular probe is used for e.g., evaluating the enhancement factor (EF). However, the EF value was not calculated and provided in this study.
- The Raman spectra for R6G molecule in Figs. 4B, 5, and 6 seem to be important for this study! However, what is the relation with the detection of trace Nicotine?
- Are both R6G and Nicotine molecules physically adsorbed on the ZnO-Tips/AuNPs surface? Is the linear relation shown in Fig. 7B the same with the plot for R6G molecule?
- fig. 9? fig. 8?
- What is the basis and applied parameters for the comparisons shown in Table 2? (the developed SERS assay for nicotine with other reported methods in the literature) Does it mean that the reported detection method can replace the traditional ones?
Author Response
Dear Sir Editor
Thank you for your email with the reviewer comments on the manuscript (biosensors-1409452). Following the reviewer’s helpful suggestions, we improved the manuscript and the queries were responded one by one. The revision was submitted for your consideration again. The kind helps were appreciated.
With best wishes
From
Dr. Haifeng Yang
Response to referee 1
In this work, ZnO tips decorated with AuNPs (ZnO-Tips/AuNPs) are fabricated to detect trace nicotine in saliva with sensitivity and low detection limit. In general, the substrate design is not special for such detection; a lot of nano-structured substrates in the literature are competent to realize the detection with similar capability. Therefore, in considering with novelty and significance, the paper is not recommended.
Thank you for your positive comment. The novelty and new insight from the composite SERS substrate have been highlighted in the revision.
- The so-called “hot-spots” was not well described. How the target species can reach the hot spots and form the SERS mechanism on the ZnO-Tips/AuNPs surface?
A: Sorry for unclear depiction in the context. As shown in Figure 1D, the abundant ZnO tips dotted with gold nanoparticle result in the aggregation of nanoparticles, which generating the great surface plasmon resonance, so-called “hot-spots”. The composite ZnO-Tips/AuNPs possessing a larger surface area with physical adsorption capability and the interaction of ZnO and pyridine moiety of Nicotine allows the target species to reach the hot spot region.
- The R6G molecular probe is used for e.g., evaluating the enhancement factor (EF). However, the EF value was not calculated and provided in this study.
A: Thank you for your suggestion. The Raman enhancement factor (EF) was evaluated according to the equation: EF = (Isurface/Isolution) × (Nsolution/Nsurface) and the EF about 1.07×108 was reached.
- The Raman spectra for R6G molecule in Figs. 4B, 5, and 6 seem to be important for this study! However, what is the relation with the detection of trace Nicotine?
A: Thank you for your comment. R6G molecule was used to evaluate the enhancement performance of ZnO-Tips/AuNPs for comparison with similar SERS substrates reported in literature.
- 4. Are both R6G and Nicotine molecules physically adsorbed on the ZnO-Tips/AuNPs surface? Is the linear relation shown in Fig. 7B the same with the plot for R6G molecule?fig. 9? 8?
A: Thank you for your comment and suggestion. R6G and Nicotine molecules physically adsorbed on the ZnO/Au surface and the pyridine moiety of Nicotine molecule could interact with ZnO. Fig. 9B shows the corresponding linear relation for Nicotine, and the linear relation for the R6G molecule has been supplemented, as shown in Fig. 6B.
Figure 6B. Calibration plot based on Raman intensity at 1363 cm−1 for R6G based on ZnO-Tips/Au substrate.
- What is the basis and applied parameters for the comparisons shown in Table 2? (the developed SERS assay for nicotine with other reported methods in the literature) Does it mean that the reported detection method can replace the traditional ones?
A: Thank you for your comment. In Table 2, the performance of SERS assay developed in this work such as linear range and limit of detection is better than other previously reported methods. Considering the development of portable Raman system, ZnO-Tips/AuNPs-based Raman assay could be used for on-site rapid test but the preparation reproducibility of ZnO-Tips/AuNPs needs to further be improved.

Reviewer 2 Report
Authors have performed the synthesis of ZnO stars and they deposited gold on the surface of those nanoparticles. Then, SERS spectrometry was performed to detect nicotine in saliva samples. I have some comments about the manuscript:
- Figure 1, they should have the same magnification in order to compare the surface structure.
- Higher magnification should be done in order to observe the gold nanoparticles of (15nm)
- It will be appreciable an EDX to know where is the gold.
- Line 121 is written “AuNPs are clearly visible in the region from 500 to 700 nm”, honestly, I have difficulties to observe the gold nanoparticles in the UV spectra. Could you explain better which corresponds with gold nanoparticles?
- Figure 5 should be described the gold nanoparticles used, SERS enhancement is different for spheres of 15nm than spheres of 60, being much higher with 60nm.
- Authors should characterize the size and shape of gold nanoparticles in order to compare, if not, it is imposible to compare the SERS enhancement.
- In the text it is not clear the effect of ZnO, which is the influence of ZnO in SERS?
- Figure 6A, is about reproducibility in the same sample, what about different SERS substrates?
- SERS is performed using 633 laser, why they did not use 785 which have higher power?
- Which is the LOD for nicotine using SERS?
- Why in table 2 is written the LOD is -12 when in the calibration authors only achieve -10?
Author Response
Dear Sir Editor
Thank you for your email with the reviewer comments on the manuscript (biosensors-1409452). Following the reviewer’s helpful suggestions, we improved the manuscript and the queries were responded one by one. The revision was submitted for your consideration again. The kind helps were appreciated.
With best wishes
From
Dr. Haifeng Yang
Response to referee 2
Authors have performed the synthesis of ZnO stars and they deposited gold on the surface of those nanoparticles. Then, SERS spectrometry was performed to detect nicotine in saliva samples. I have some comments about the manuscript:
- Figure 1, they should have the same magnification in order to compare the surface structure. Higher magnification should be done in order to observe the gold nanoparticles of (15nm). It will be appreciable an EDX to know where is the gold.
A: Thank you for your suggestion. The magnification of Figure 1(C) is to present the whole morphology of a bunch flower-like ZnO, and in Figure 1(D) the gold nanoparticles with average size of 40 nm could be observable. Following your suggestion, the EDS mapping is added in Figure 2 and the energy-dispersive X-ray spectroscopy (EDS) result in Figure A1, confirming the presence of AuNPs in the composite substrate and AuNPs are evenly distributed on ZnO.
Figure 2. EDS mapping of ZnO/Au.
Figure A1. EDS result of ZnO/Au.
- 2. Line 121 is written “AuNPs are clearly visible in the region from 500 to 700 nm”, honestly, I have difficulties to observe the gold nanoparticles in the UV spectra. Could you explain better which corresponds with gold nanoparticles?
A: Thank you for your pointing out the mistake. SPR band of AuNPs could be visible in the region from 500 to 600 nm. The UV-vis spectrum of ZnO/Au2 is separately shown in Figure A2.
Figure A2. UV-vis spectrum of ZnO/Au.
- Figure 5 should be described the gold nanoparticles used, SERS enhancement is different for spheres of 15nm than spheres of 60, being much higher with 60nm.
Authors should characterize the size and shape of gold nanoparticles in order to compare, if not, it is impossible to compare the SERS enhancement.
A: Thank you for your comment. The SERS performance of gold nanoparticles of different sizes reduced on ZnO has been compared by optimizing the volume of reducing agent added in Figure 5. The diameter of gold nanoparticles was evaluated according to the equation:and the diameters of gold nanoparticles are about 71.5, 43 and 15 nm, respectively. Clearly, the ZnO-Tips dotted with 43 nm AuNPs exhibited the highest SERS activity.
- 4. In the text it is not clear the effect of ZnO, which is the influence of ZnO in SERS?
A: Thank you for your comment. For checking enhancement effect for ZnO tips, SERS spectra of R6G were acquired on ZnO-Tips/Au and pure AuNPs, respectively. Obviously in Figure 7, SERS effect for R6G recorded on ZnO-Tips/Au is much greater than that of gold nanoparticles. It depicts that the ZnO tips improve the SPR enhancement effect of AuNPs through effective aggregation of nanoparticles and mutual interaction of ZnO and AuNPs to optimize the electronic structure of composite SERS substrate. In addition, the chemical enhancement effect of ZnO also contributes the SERS detection sensitivity. The reason is that the work function of ZnO (5.2 eV) is larger than that of Au (5.1 eV) and the Fermi energy level of ZnO is lower than that of Au, resulting in electron transfer from AuNPs to ZnO occurs until their level of Fermi energy attains equilibration. As a result, it is beneficial to improvement of chemical interaction between target molecules and ZnO. Simultaneously, it is conducive to more molecules closely approaching to the vicinity of hot spots region to further provide enhancement effect.
- 5. Figure 6A, is about reproducibility in the same sample, what about different SERS substrates?
A: Sorry for the unclear statement. In the experimental section, three different substrates were selected for examining preparation reproducibility. For each substrate, 5 points in the coffee ring area were randomly selected for evaluating the relative standard derivation of detections.
- 6. SERS is performed using 633 laser, why they did not use 785 which have higher power?
Which is the LOD for nicotine using SERS?
Why in table 2 is written the LOD is -12 when in the calibration authors only achieve -10?
A: Thank you for your comment. SPR band of AuNPs locating in the region from 500 to 600 nm shows that 633 laser line is suitable for SERS experiments by using ZnO-Tips/AnNPs.
The LOD for nicotine has been given in the manuscript according to the formula of LOD=3 suggested by the IUPAC.
where the RSD value is the standard deviation for repeating the same experiment and the BEC value is the absolute value of the intercept between the linear regression equation and the x-axis.
Based on the results in Figure 7B, the specific calculation for LOD is obtained as follows:
LOD1589cm-1=3*10-10.25 mol/L=8.9*10-12 mol/L
I am sorry that the 10-10 mol/L in the text is written wrongly. The calculation of LOD has been given in the revision.

Reviewer 3 Report
The manuscript submitted to Biosensors entitled "ZnO-TiPs Dotted with Au Nanoparticles Advanced SERS Determination of Trace Nicotine" by Jiaying Cao et al.. Herein, zinc oxide (ZnO) tips decorated with gold nanoparticles (AuNPs) are fabricated and used for surface-enhanced Raman scattering (SERS) detection of nicotine. The clear majority of relevant literature about this subject is referenced and adequately discussed by the authors. Nicotine detection is a relevant analyte with an increasing significance in recent years, and novel methods for its accurate detection are a welcome addition to the literature.
However, a few doubts should be clarified before publication. For instance:
a) In the experimental section, there is a general procedure for synthesising the AuNPs/ZnO. However, in the results and discussion, the authors discuss the optimisation preparation where different quantities of trisodium citrate solution are added. The authors should describe all the procedures for the materials studied in this report.
b) An initial assessment of the manuscript is difficult to understand the materials denominations in the optimisation section where it seems that ZnO/Au2 is the exact same thing as ZnO-Tips/AuNPs. The authors should uniformise the names of the different materials. Two names for the same material is somehow confusing.
Author Response
Dear Sir Editor
Thank you for your email with the reviewer comments on the manuscript (biosensors-1409452). Following the reviewer’s helpful suggestions, we improved the manuscript and the queries were responded one by one. The revision was submitted for your consideration again. The kind helps were appreciated.
With best wishes
From
Dr. Haifeng Yang
Response to referee 3
Comments and Suggestions for Authors
The manuscript submitted to Biosensors entitled "ZnO-TiPs Dotted with Au Nanoparticles Advanced SERS Determination of Trace Nicotine" by Jiaying Cao et al.. Herein, zinc oxide (ZnO) tips decorated with gold nanoparticles (AuNPs) are fabricated and used for surface-enhanced Raman scattering (SERS) detection of nicotine. The clear majority of relevant literature about this subject is referenced and adequately discussed by the authors. Nicotine detection is a relevant analyte with an increasing significance in recent years, and novel methods for its accurate detection are a welcome addition to the literature.
Thank you for your positive comment.
However, a few doubts should be clarified before publication. For instance:
- a) In the experimental section, there is a general procedure for synthesising the AuNPs/ZnO. However, in the results and discussion, the authors discuss the optimisation preparation where different quantities of trisodium citrate solution are added. The authors should describe all the procedures for the materials studied in this report.
A: Thank you for your suggestion. All the procedures for the materials studied have been described in revision.
- b) An initial assessment of the manuscript is difficult to understand the materials denominations in the optimisation section where it seems that ZnO/Au2 is the exact same thing as ZnO-Tips/AuNPs. The authors should uniformise the names of the different materials. Two names for the same material is somehow confusing.
A: Thank you for your suggestion. The names of the different materials have been unified.

Round 2
Reviewer 1 Report
From the authors’ responses, two replies should be furthermore clarified:
1. In the previous questions 3 and 4, the authors should discuss if the target Nicotine molecules are physically or chemically adsorbed on the ZnO-Tips/AuNPs surface?
Moreover, in the revised manuscript (page 11, lines 199201)- “In Figure 9A, the SERS intensity of the nicotine increases as the nicotine concentration increasing. The SERS band at 1589 cm−1 corresponding to the stretching of the pyridine ring of nicotine was selected for quantitative analysis.”: However, there is no reference to show that the peak at 1589 cm-1 is the characteristic structure for the Nicotine molecule that is particularly suitable for the concentration-dependent studies.
2. In the previous question 5, the authors should discuss more about the comparisons shown in Table 2 (page 13, lines 223-226). There are plenty of SERS-active substrates in the literature, why the studied one, to be included in a diagnostic device, is much competitive?
Author Response
Dear Sir Editor
Thank you for your email with the reviewer comments on the manuscript (biosensors-1409452). Following the reviewer’s helpful suggestions, we improved the manuscript and the queries. The revision was submitted for your consideration again. The kind helps were appreciated.
With best wishes
From
Dr. Haifeng Yang
- In the previous questions 3 and 4, the authors should discuss if the target Nicotine molecules are physically or chemically adsorbed on the ZnO-Tips/AuNPs surface?
Moreover, in the revised manuscript (page 11, lines 199201)- “In Figure 9A, the SERS intensity of the nicotine increases as the nicotine concentration increasing. The SERS band at 1589 cm-1 corresponding to the stretching of the pyridine ring of nicotine was selected for quantitative analysis.”: However, there is no reference to show that the peak at 1589 cm-1 is the characteristic structure for the Nicotine molecule that is particularly suitable for the concentration-dependent studies.
A: Thank you for your comment. Compared with the normal Raman bands acquired from nicotine reported in literature [1], the SERS bands of nicotine recorded on ZnO-Tips/Au substate happened bule-shifts, which hinting the adsorption of nicotine on the substate surface in physical and chemical fashions. The SERS band at 1589 cm-1 may be assigned to stretching of the pyridine ring of nicotine according to the previously reported work [1]. Following your suggestion, the quantitative relationship was also complemented based on the characteristic peak of nicotine at 1031 cm-1 and a linear concentration dynamic range locates from 10-10-10-6 mol/L with a correlation coefficient (R2) of 0.9663. Clearly, the SERS band at 1589 cm-1 is observable without any spectral interferences.
Figure A1. Calibration plot based on Raman intensity at 1031 cm−1 for Nicotine based on ZnO-Tips/Au substrate.
- In the previous question 5, the authors should discuss more about the comparisons shown in Table 2 (page 13, lines 223-226). There are plenty of SERS-active substrates in the literature, why the studied one, to be included in a diagnostic device, is much competitive?
A: Thank you for your suggestion. The comparison with other SERS substrates assays for nicotine have been added in Table 2. In this work, the proposed ZnO-Tips/Au-based SERS assay exhibits superior sensitivity and reasonable quantitative analysis performance due to the improved AuNPs SPR effect via interaction with ZnO tips. What’s more, ZnO-Tips/Au substrate has good stability under ambient condition, which is beneficial to real application.
Reference
- Barber, T. E., List, M. S., Haas, J. W., & Wachter, E. A. (1994). Determination of Nicotine by Surface-Enhanced Raman Scattering (SERS). Applied Spectroscopy, 48(11), 1423–1427.

Reviewer 2 Report
Dear authors,
Thank you for addressing all my comments, in my point of view the manuscript has been improved.
Author Response
Thank you for your positive comment.
Round 3
Reviewer 1 Report
In the revised manuscript, the authors did not reply the points:
- (Questions 1-4) The R6G probe molecule was used to confirm that the substrate exhibited SERS-active. It should not be presented as an important results in this study. In addition, the EF and other calculations should be managed in the materials and methods.
- (Question 5) The major concern is the peaks assignment of SERS spectrum for the trace nicotine in saliva. In the manuscript, it was not identified by the peaks assignment in the spectrum or shown the in a e.g., Table! The identical data should be provided!
Author Response
Dear Sir Editor
Thank you for your email with the reviewer comments on the manuscript (biosensors-1409452). Following the reviewer’s suggestions, we improved the manuscript and the queries were answered one by one. The revision was submitted for your consideration again. The kind helps were appreciated.
With best wishes
From
Dr. Haifeng Yang
- (Questions 1-4) The R6G probe molecule was used to confirm that the substrate exhibited SERS-active. It should not be presented as an important results in this study. In addition, the EF and other calculations should be managed in the materials and methods.
A: Thank you for your comment. The original Figures 5 and 6 for SERS spectra of R6G on the substrates were moved into the supplementary material and the EF and other calculations were also addressed in supplementary material.
- (Question 5) The major concern is the peaks assignment of SERS spectrum for the trace nicotine in saliva. In the manuscript, it was not identified by the peaks assignment in the spectrum or shown the in a e.g., Table! The identical data should be provided!
A: Thank you for your suggestion. The assignments of the main SERS bands for nicotine in Figure 8 have been marked and tabulated in Table S1.